# Open banking: A bibliometric analysis-driven definition

**Gorka Koldobika Briones de Araluze**[1]*, **Natalia Cassinello Plaza**[2]

**1** Universidad Pontificia Comillas, Madrid, Spain, **2** Departament of Financial Management, Facultad de Ciencias Económicas y Empresariales, Universidad Pontificia Comillas, Madrid, Spain

☯ These authors contributed equally to this work.

* gbrionesde@comillas.edu

## Abstract

"Open banking," as a concept, was initially developed by a UK regulation to foster competition in banking through sharing client data (with their consent) amongst competitors. Today, it is regulated in several most relevant banking jurisdictions. Despite its growing relevance, consensus about the definition of open banking is lacking. This study examines 282 articles on open banking using bibliometric clustering techniques. Moreover, within the 282 articles and applying discourse analysis, we analyze 47 idiosyncratic definitions of open banking to test an integral framework that supports our proposed definition of the concept. Our study contributes to the literature by providing a generalized multidisciplinary definition of open banking. It identifies four main drivers behind the concept: business model change, client data sharing, incorporation of technological companies (fintechs and others), and regulation. These four elements, which should be considered in new regulations in the globalized banking sector, foresee open banking as a critical enabler of a new strategic dynamic in banking.

**Data Availability Statement:** All relevant data are within the manuscript and its Supporting Information files.

**Funding:** The authors received no specific funding for this work.

## 1. Introduction

What is open banking? Since the inception of the "Open Banking Working Group" in the United Kingdom in 2015, open banking has generally been considered as the platformization of the retail banking industry [1, 2]. To date, it has spread worldwide from the UK to Continental Europe, America, and Asia, constituting one of the retail banking industry's shaping forces of the future [3, 4]. Thus, on top of the open banking initiative in the UK and PSD2 (Payment Services Directive 2) in the European Union, there are open banking regulations in Australia, India, México, and Brazil, and forthcoming regulations in Russia and Canada.

The essence of open banking regulations is to recognize the banking clients' right to share their transactional data with authorized third parties and detailed provisions on how to materialize this right [5]. Despite its apparent simplicity, this data-sharing right constitutes the primary vector for fostering the transformation of the retail banking sector from a closed business model to an open platform, similar to what occurred in telecommunications, power, and gas industries [6].

**Competing interests:** The authors have declared that no competing interests exist.

Open banking originated from practitioners and was inspired by the open data, open-APIs (Application Programming Interfaces), and open innovation philosophies [7] applied to the retail banking business [8, 9]. The business community is analyzing this phenomenon extensively, understanding it as a "*collaborative model in which banking data is shared through APIs between two or more unaffiliated parties to deliver enhanced capabilities to the marketplace*" [10].

Its first implementation worldwide materialized in the UK. It was requested by the Competition and Markets Authority as a foundational strategy to ascertain that personal current accounts, as well as small and medium-sized enterprises' banking markets, serve customers better. This issue emanated from a retail banking market investigation concluded in 2016 [8]. It also inspired the European Commission to publish the PSD2 [7, 11, 12]. Although open banking is still in its initial stages of development, the concept has been embraced by practitioners and regulators, being regarded as one of the shaping forces of the financial industry worldwide [4].

Nevertheless, despite existing literature acknowledging the importance of open banking as a critical retail banking industry's transformational lever [13], open banking as a research object still lacks conceptualization both theoretically and empirically [14]. Academic literature on the subject is still in its early stages of development. Out of 990 documents registered in the Google Scholar database (Aug 6, 2021) containing the term "open banking," only 57 were published in Scopus-rated peer-reviewed academic journals.

Considering its international and multidisciplinary nature, open banking as a research object presents several challenges. To begin with, open banking is being studied in many academic fields, and researchers who represent different disciplines seem not to converge on a shared definition of open banking [14]. Additionally, most authors researching the topic leverage idiosyncratic definitions aligned with their respective research focus [15]. Moreover, subtle differences among open banking regulations worldwide create confusion when comparing publications from different geographies [3]. Hence, our study aims to establish a generalized definition of open banking and its varying interpretations in different disciplines and geographies. A generalized definition of open banking would add consistency and robustness to existing research, laying out a solid foundation to support high-quality research on the phenomenon.

Apart from a generalized definition, understanding different contexts in which the term "open banking" is used is also essential. Open banking can be discussed from different perspectives (regulatory, technological, economic, and managerial) that imply different nuances, which should be identified. Additionally, it is also critical to validate a generalized definition under these different contexts to assure that it works properly in all of them.

This study aims to understand the contexts and meanings of the term "open banking" and proposes a generalized definition that can be used unambiguously in the academic literature. For this objective, two methodologies are used. First, through clustering-based bibliometric analysis, 282 academic articles are analyzed to identify the areas, contexts, or meanings of "open banking." Second, applying a "discourse analysis" methodology, the 47 definitions of open banking found in the literature are examined, and a generalized definition of the term applicable to all open banking connotations is proposed.

Our study makes several contributions to the literature. First, it performs a review of the pre-existing literature on open banking applying bibliometric techniques. Second, a generalized definition of open banking and its four applications (business model, fintech, data-sharing, and regulation) are proposed. Third, the 47 existing open banking definitions are systematically analyzed, and a classification is proposed for them (institutional, ecosystem, and client). Likewise, generated inductively, an "open banking integrated definition framework" is formulated based on eight elements that can be applied to similar definitions. Finally, the

Hirschman Herfindahl Index (HHI) is used innovatively within the discourse analysis to measure the degree of consensus regarding the definition.

## 2. Literature review and research question

Open banking is a new phenomenon in the banking industry and an even newer concept in academia. Before 2016, only four articles contained the term "open banking" in academic or grey journals. Hence, open banking can be considered a new study object.

Existing literature can be grouped into three blocks: regulatory, technical, and managerial. The regulatory literature analyzes the legislation that supports open banking (European Union's Second Payment Services Directive [PSD2], UK's Open Banking Standard, Australia's Consumer Data Right, Singapore's Personal Data Protection Act, India's Aadhaar and Unified Payments Interface, and similar regulatory pieces being analyzed and approved in Hong Kong, Canada, Brazil, [BCB Circular No. 4,015/2020], and Mexico (Ley Fintech). Existing publications either focus on a single jurisdiction [16–19] or compare different legislations [20, 21]. From a technology perspective, existing literature focuses on the underlying infrastructure [22–24] as well as on the acceptance of the open banking technology from the customer's perspective [25–27]. Managerial literature analyzes structural changes in the demand and supply of financial services in the retail banking market due to open banking [7, 10, 12, 28–30]. Finally, other fields such as microeconomics are also starting to analyze the phenomenon [31].

Nevertheless, despite a growing academic interest in open banking, foundational literature is still missing. There are no publications analyzing the origins of open banking (why open banking is needed), the nature of the phenomenon (how open banking has developed in different geographies) or, even more basic, what open banking is. As a matter of fact, there are only three publications devoted to establishing a definition of open banking. van Zeeland and Pierson (2021) follow a bibliometric and discourse analysis approach for open banking, but they fail to propose a definition, concluding that:

> *"Open Banking could be all kinds of things, from a remedy to an ecosystem, or most often: a (business) model of some sort. Its purposes are considered to be providing new ('better', 'customer-centric') services to customers and improving competition in the banking market by letting 'third parties' in."* [14]

O'Leary et al. (2021), building on an open data lenses approach, propose the following definition:

> *"An initiative which facilitates the secure sharing of account data with licensed third parties through Application Programming Interfaces (APIs), empowering customers with ownership of their own data. The initiative aims to increase competition in retail banking by developing innovative products and services which will bring increased value to customers."* [15]

Finally, Laplante and Kshetri (2021) approach the need for a definition of open banking, but do not provide a generalized definition other than describing the phenomenon as:

> *"Open banking describes a special kind of financial ecosystem. The ecosystem provides third-party financial service providers open access to consumer banking, transaction, and other financial data from banks and nonbank financial institutions through the use of application programming interfaces (APIs)."* [32]

The existing definitions of open banking present three types of problems fundamentally: perspective bias, discipline bias, and purpose bias. Starting with the perspective bias problem, open banking is a tripartite scheme between the owner of the data, custodian, and third party who accesses it. Any general definition must consider the three agents to avoid partial or incomplete analysis of the phenomenon. Regarding the discipline bias problem, researchers tend to confuse the context in which open banking is used in their discipline with a generally applicable definition. Thus, technical literature focuses exclusively on the technological support of the phenomenon, the regulatory literature on its legal support, and the management literature on the possible implications for the business model. However, a generalized concept of open banking must be able to encompass all its contexts of use and not just one of the meanings. Finally, the purpose bias problem consists of giving open banking a specific purpose other than the one for which it was formulated: to increase competition in retail banking by facilitating the entry of new competitors. Considering the combined effect of the three biases, the definitions proposed so far of open banking do not allow the construction of solid and generalizable knowledge about the phenomenon, which is a significant caveat on its development.

One last question is why academic research on open banking is relevant. There are no global figures for the investment required to materialize open banking. According to Tink, one of the world's leading open banking service providers [33], the average open banking expenditure for a retail bank in Europe in 2020 was €83.1 Mn. So, the aggregated figure for the system should be in the range of tenths of billions annually, just for Europe. Nevertheless, we have no evidence, based on scientific studies, of the intention of customers to use services based on open banking. There is no scientific evidence on how open banking can impact value creation and distribution in retail banking. No robust academic studies explain the conditions under which customers are willing to share data with third-party providers. In short, the academia has dealt with accessory elements of open banking but not with the central aspects of the phenomenon. The lack of a robust and generally shared definition of the phenomenon allowing collaboration among researchers and a holistic view of the phenomenon, is at the heart of this knowledge gap.

Thus, a generalized definition of open banking together with a detailed understanding of different contexts in which the "open banking" concept is used is a relevant gap in the academic literature that needs to be filled. A particular contribution of this study is that it tackles the research question through a multidisciplinary approach, integrating views from different knowledge domains and through mixed quantitative-qualitative techniques, specifically bibliometric research and discourse analysis.

## 3. Methodology

This study follows a three-tiered approach to present a potential generalized definition of open banking (Fig 1). First, using bibliometric techniques, we map existing literature (282 documents) and, by applying co-word analysis, cluster co-occurring terms to identify conceptual domains related to open banking. The clustering analysis is executed using *Visualization of Similarities* (VoS), an evolution of Multidimensional Scaling (MDS) algorithms. From this analysis, we identify four clusters that inform the existing open banking literature and examine the interaction among them. Second, by applying a discourse analysis approach, we analyze existing definitions of open banking in the literature (47 definitions found in the 282 articles) to reveal critical attributes mentioned in these definitions considering their disciplinary and geographical variations. We, then, profile the descriptors used concerning each attribute and propose a framework to analyze existing open banking definitions. Third, based on the

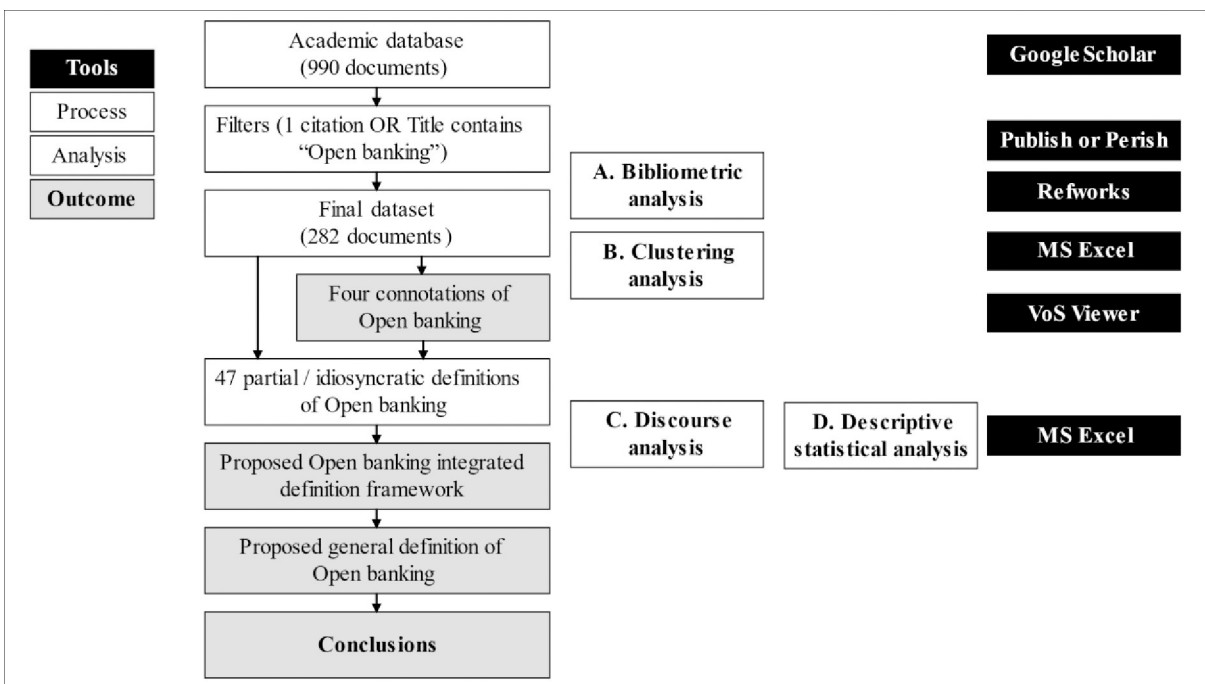

**Fig 1. Overview of the process.**

analysis, we outline an integrative definition of open banking, identify limitations of the investigation, and propose future research developments.

The analysis supporting this publication combines two methodological approaches: bibliometric and discourse analysis. First, we identify and analyze all relevant open banking literature and cluster the main perspectives on the topic by leveraging bibliometric techniques. Then, we extract 47 idiosyncratic, partial, or working definitions of open banking identified in the dataset. Applying critical discourse analysis, a method that has been accepted in the academic literature as a valid procedure for social sciences research [32, 34], we systematically examine the 47 definitions to deduce a general definition for open banking and interpret the results.

## 3.1. Bibliometric analysis

**3.1.1. Analytical approach.** Bibliometrics refers to the field that investigates groups of publications applying quantitative analysis methods [35]. Although this technique was initiated during the 1950–1960 period, it gained traction in the last two decades with the emergence of large electronic databases of academic articles, such as Web of Science (WoS) and Scopus, and the generalization of bibliometric analytics software packages, such as Gephi, Leximancer, and VOSviewer [36].

Bibliometric analysis techniques can be divided into three prominent families according to their goal [37]: techniques for establishing a relationship between authors (co-author analysis), techniques that aim at establishing a relationship between publications (citation analysis, co-citation analysis, and bibliographic coupling), and techniques for defining relationships within the content of selected publications (co-word analysis). Considering the relative novelty of the topic under consideration and the lack of consolidation of the academic sources considered,

this study focuses on co-word analysis to identify the underlying constructs of the open banking concept.

From an analytical point of view, core techniques of bibliometric analysis can be divided into performance analysis and science mapping [37]. As an evolution of science mapping core techniques, enrichment techniques allow outcome augmentation to produce more advanced insights. This study applies clustering and visualization, both enrichment techniques, to perform a co-word analysis on the dataset that comprises all relevant open banking academic literature. Co-word analysis clustering and visualization techniques' output is a network of topics and their associations, which represent the conceptual domain of a research field. Although clustering and visualization techniques are conceptually different, they usually go hand in hand [37]. In this study, they are applied simultaneously to analyze the dataset.

**3.1.2. Dataset building and process.** Although the first open banking regulation was approved in 2017 in the UK, the concept's origins are uncertain. Simon Redfern founded the Open Bank Project in 2012 [38]. But even before that, academic articles have been containing references to "open finance" and "financial aggregation" since 2002 [39]. Consequently, our database includes articles about "open banking" since 2002.

The initial dataset consists of 990 documents identified through a search in the Google Scholar database for articles using "open banking" as a keyword, conducted on August 6, 2021. The search is carried out through the Publish or Perish software tool.

Since its launch in 2004, Google Scholar has positioned itself as the most comprehensive academic citations database compared with alternative options such as WoS or Scopus, especially for humanities and social sciences [40]. However, Google Scholar contains articles not published in peer-reviewed journals, which requires additional filtering to ensure the quality of the database. Thus, Publish or Perish is commonly used in bibliometric analysis to filter academic publications databases [40].

Only documents written in English are selected due to the clustering analysis' language requirements (663 articles). Two filters are subsequently applied: documents containing "open banking" in the title (92 papers) and records that contained "open banking" in the abstract and that had at least one citation (264 documents), obtaining 356 articles. To include articles with at least one citation is a potential quality filter of literature referenced in Google Scholar and is consistent with academic procedures [41, 42] and recent bibliometric publications on the topic [14]. An additional check is performed to ensure that all the articles referenced in Scopus and WoS related to the topic are contained in the filtered database. After that, the remaining papers are fully read with two objectives. First, on the bibliometric side, to reject false positives of the combination of the words "open" and "banking," obtaining the final list of 282 documents from 2002 to 2021 (Fig 2). The resulting dataset is uploaded to RefWorks, a commonly used reference manager software [26]. Second, on the content analysis approach, to extract all the definitions of "open banking" included in the dataset. Forty-eight definitions of "open banking," transcribed in Tables 3–5 of S1 Annex, are identified and recorded in an excel database (S1 Annex) [42].

Due to limitations in obtaining full-text searchable versions of all the articles in the dataset, co-wording analysis is performed only on the titles and abstracts. This approach is consistent with existing bibliometric techniques as described in the literature [36]. These 282 articles yield 5,000 terms; out of which only those with five or more occurrences are selected (377). Ten generic terms (article, case, case study, chapter, example, interview, number, paper, study, and year) are removed from the selection, finishing with 367 terms. These terms are clustered, defining a minimum size of 25 items per cluster to avoid micro fragmentation of clusters. This process results in four clusters discussed in the results section. The normalization method applied is Linear / Logarithmic, and the proposed visualization layer is built using an attraction

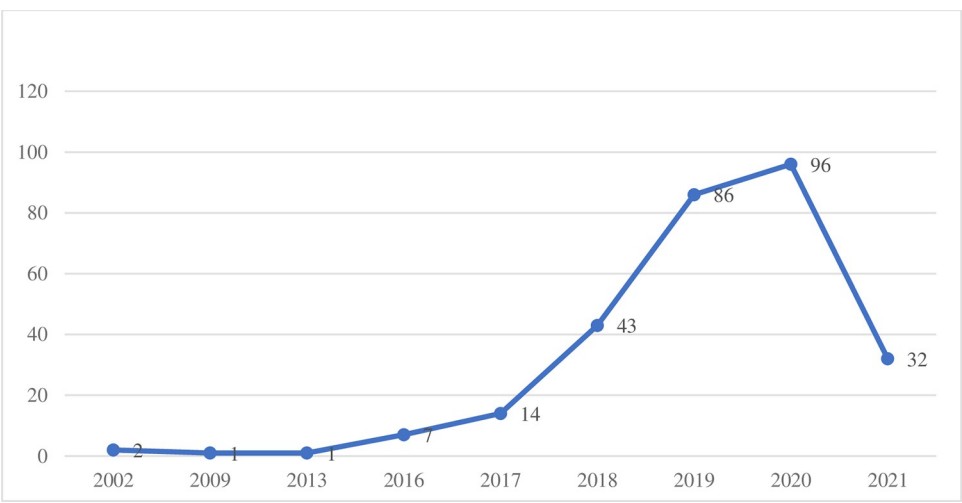

**Fig 2. Final dataset publications per year.**

parameter of 3 and a rejection parameter of 0. The minimum cluster size is set at 25 [43], and the iterations number is set at 50.

## 3.2. Discourse analysis

During the bibliometric analysis dataset-building process, 47 definitions of "open banking" are identified. Each one of them appears in just one article. Although only three articles [14, 15, 32] are devoted to defining open banking, most articles dealing with the topic leveraged idiosyncratic or working definitions. The definitions are extracted and systematically analyzed from two perspectives.

First, a semantic approach is used to understand the role of each definition component. Eight semantic/grammatical elements are identified by applying an inductive approach: Nature, Consent, Subject, Action, Object, Recipient, Process, and Purpose. These eight elements constitute our proposal of an "open banking integrated definition framework," which is discussed in detail in the Results section.

Second, to test the framework's robustness, a descriptive statistics approach is applied to understand (i) the degree of completion of the definitions identified according to the proposed framework and (ii) the level of convergence/dispersion in the definitions. HHI is applied to the definitions to assess the convergence/dispersion within each element.

HHI is a well-established measure, often used in economics to analyze the degree of concentration of a given market. It is calculated according to the following expression [44]:

$$HHI = \sum_{1}^{n} c_i^2,$$ [1]

where ci accounts for the (market) share of the -I element and where

$$\sum_{1}^{n} c_i = 100$$ [2]

In our case, we calculate HHI for each conceptual field identified in the definitions. For each of the eight elements, if the 47 definitions used the same concept, HHI would yield a 10,000 (maximum value). If different concepts were used by the 47 definitions, HHI would be 212.8 [47 x $(100/47)^2$].

## 4. Results

### 4.1. Bibliometric analysis and main research trends

As previously mentioned, open banking is a relatively new term in academic literature. The first time it appeared in academic literature fully aligned with the current interpretation was in 2009, but it started to take-off after 2016. The data for 2020 and 2021 (Fig 2) might be affected by the criteria of choosing auxiliary publications that were cited at least once.

Regarding the nature of the documents, the dataset is highly heterogeneous: 20.2% documents [57] are articles published in Scopus rated journals; 5.0% [14] are Scopus-listed conference proceedings, and the remaining 211 are primarily reports, books or book sections, and academic dissertations (Fig 3).

It is worth noting that despite the limited academic relevance of existing literature, it is evolving toward more journal publications and Scopus-listed conference proceedings, implying higher relevance within the academic community (Fig 4).

Although the main field of study for open banking, following Scopus classification, is *Business*, *Management*, *and Accounting*, interest in the phenomenon is growing in other disciplines, too. In fact, in 2020, *Business*, *Management*, *and Accounting* accounted for 30.2% of the documents published, *Computer Sciences* accounted for 27.1%, *Social Science–Law* accounted for 14.6%, *Economics*, *Econometrics*, *and Finance* accounted for 11.5%, and other fields (*Medicine*, *Engineering*, *Social Science–Other*) accounted for 16.7% (Fig 5).

**Observation 1.1.** While the interest of academia in the open banking phenomenon is still limited, it is growing significantly over the last few years.

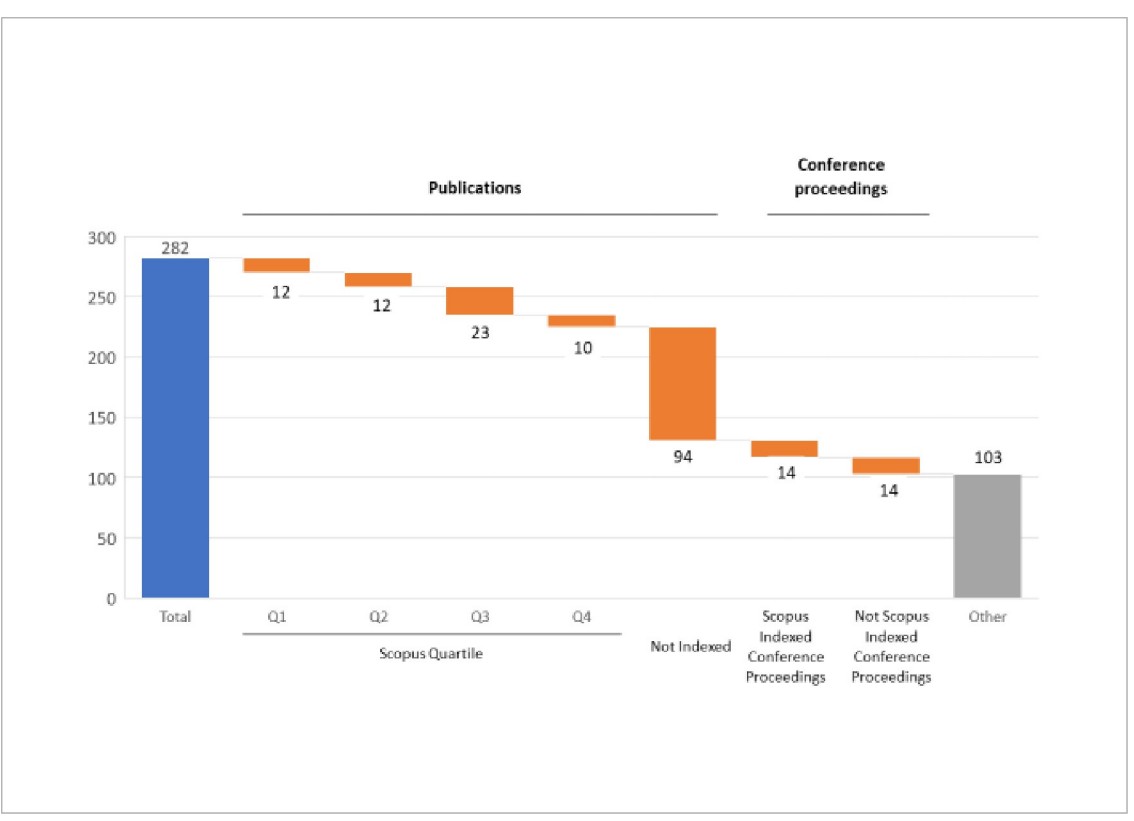

**Fig 3. Dataset classification by nature.**

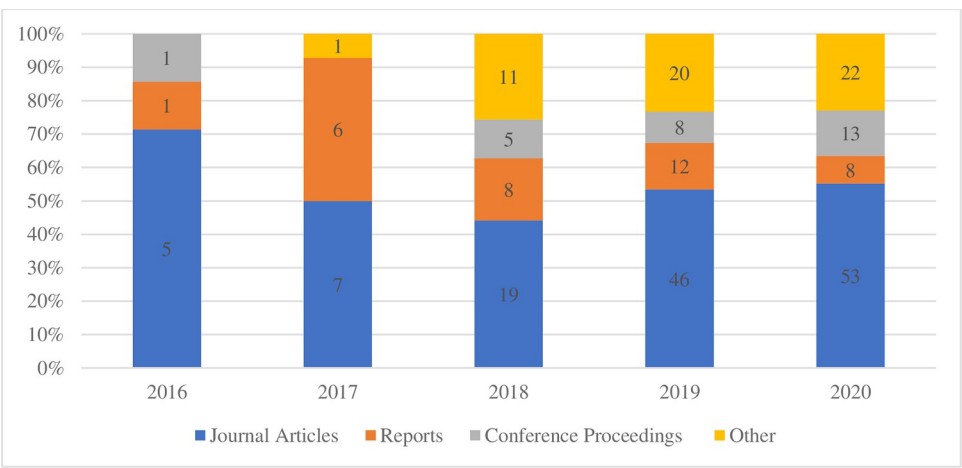

**Fig 4. Evolution of documents in the dataset by category (2016–2020).**

**Observation 1.2** The quality of academic literature analyzing open banking is increasing, with a higher number and proportion of publications in higher-rated magazines.

**Observation 1.3** Open banking is a multidisciplinary phenomenon that is being studied by several disciplines.

## 4.2. Clustering analysis and main conceptual domains (drivers) of open banking

Through the application of the VoS algorithm, four clusters are identified (Fig 6). These clusters are groups of keywords that appear in at least five documents. Table 1 summarizes the top 10 keywords for each cluster.

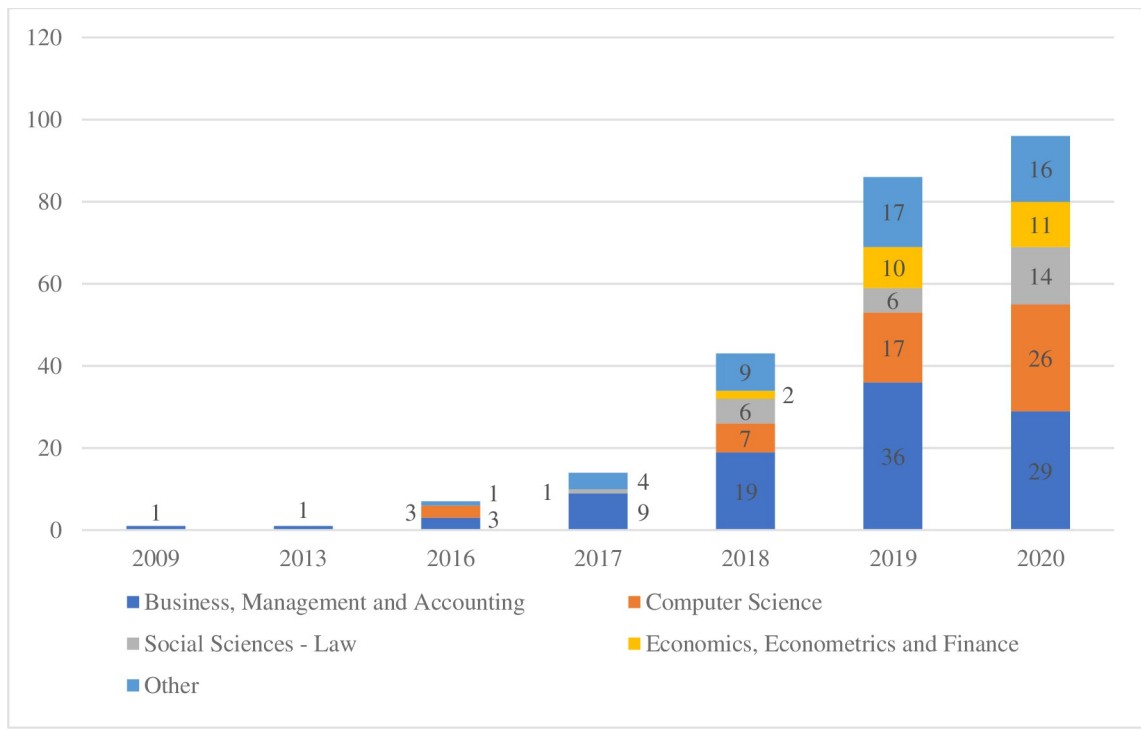

**Fig 5. Final dataset documents by category (2009–2020).**

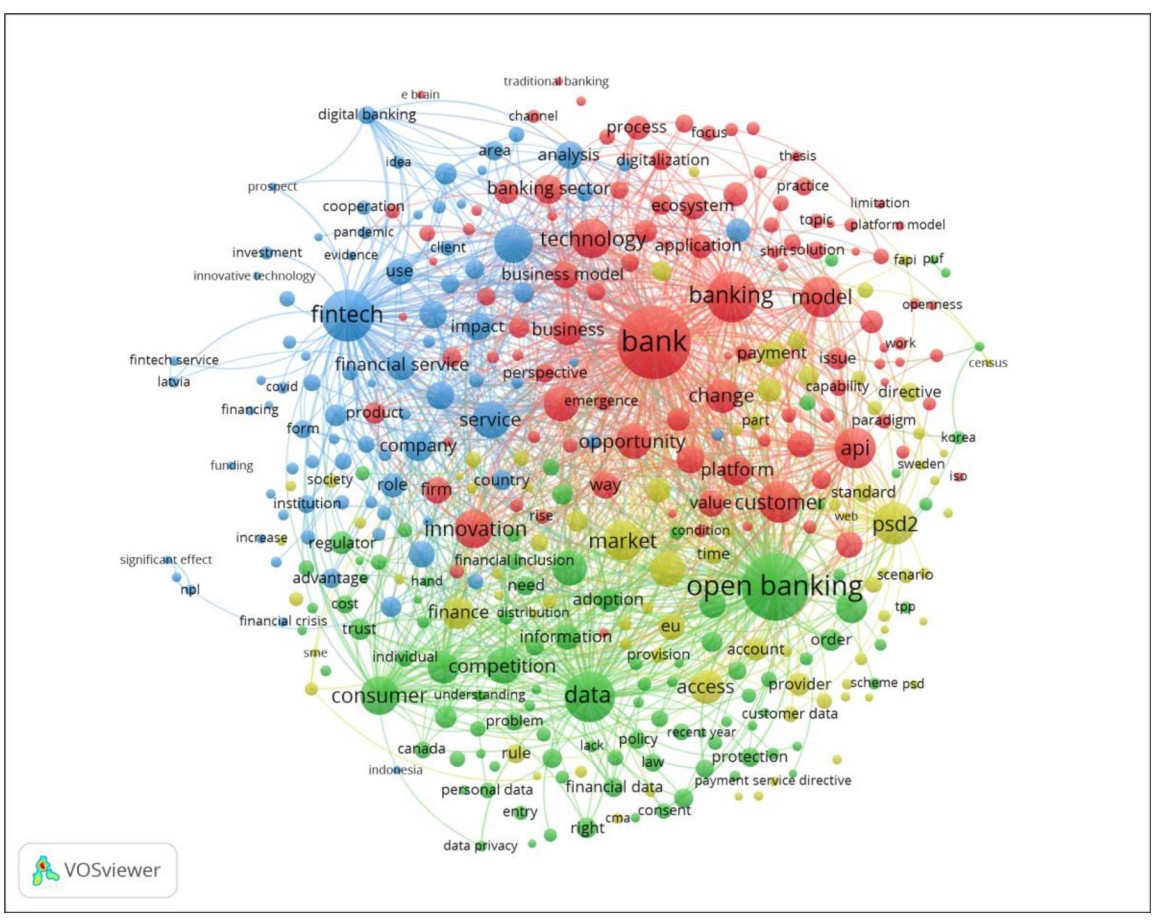

**Fig 6. Graphical cluster representation.**

Before coding, both researchers agreed on the coding method: based on heuristics, assigning to the cluster a description that explained at least 50% terms included in each cluster. Both researchers performed independent coding, and the results were compared and discussed to obtain the proposed interpretation.

**Table 1. Main components by cluster.**

| Rank | Cluster 1 (Bus. Model Platform) | Cluster 2 (Data sharing) | Cluster 3 (Fintech) | Cluster 4 (Regulation) |
|------|--------------------------------|--------------------------|---------------------|------------------------|
|      | (Red) | (Green) | (Blue) | (Yellow) |
| 1 | bank | open banking | fintech | psd2 |
| 2 | customer | data | development | market |
| 3 | model | consumer | company | regulation |
| 4 | API | competition | financial service | access |
| 5 | technology | challenge | economy | finance |
| 6 | innovation | risk | world | EU |
| 7 | opportunity | framework | use | payment |
| 8 | industry | system | banking service | transaction |
| 9 | change | information | country | future |
| 10 | platform | adoption | implementation | account |

*Cluster 1* (*Business model platformization*): the initial list included both "bank" and "banking," and both terms were consolidated. Here, open banking could be interpreted as the transformation process of the retail banking business model toward a platform leveraging API technology and fostering innovation.

*Cluster 2* (*Data sharing*): summarizes the main open banking features: a new framework involving data (information) sharing and opening the banking market to competition, which poses new challenges and risks for legacy players.

*Cluster 3* (*Fintech*): summarizes the ecosystem impact of the fintech phenomenon as a new competitor for financial institutions. From the initial outcome of the analysis, several generic keywords were removed for interpretation purposes: "research," "impact," "use," "level," "role," "factor," and "effect." Additionally, "service" was consolidated with "financial services" for clarity.

*Cluster 4* (*Regulation*): reflects the regulatory side, focusing on the legal and jurisdictional implications.

**Observation 2.1.** Open banking as a research field is built on four domains: business model platformization, data sharing, fintech, and regulation, all of which can be interpreted as different connotations of open banking.

**Observation 2.2.** Each identified cluster has a strong relationship with different knowledge domains.

**Observation 2.3.** Clustering analysis confirms the adequacy of a multidisciplinary approach, considering the heterogeneous nature of the phenomenon and the associated literature.

## 4.3. Analysis of open banking definitions

Next, the final 282-document dataset was manually read, searching for formal or idiosyncratic definitions of open banking, the result of which is 47 definitions (S1 Annex; Tables 3–5)

Existing literature does not provide a framework to analyze "open banking" or similar definitions. Following similar approaches in the academic literature [45, 46], the authors proceed to build an ad-hoc framework: the "open banking integrated definition framework" based on induction from the 47 existing definitions. This process identifies eight elements in which all current definitions can be decomposed.

The definitions are then decomposed into eight elements categorized into the following three blocks and analyzed to deduce a general definition of open banking constituting the "open banking integrated definition framework": (i) Conceptual elements: Nature (*How can the phenomenon be classified*?) and Consent (*What is the enforceability*?), (ii) Core attributes: Subject: (*Who is the actor*?); Action (*What is expected from the Subject*?); Object (*What is the target of the Action*); Recipient (*Who is affected by the Action*?) and Process (*How does the Subject interact with the Object and with the Recipient*?), (iii) Purpose (*What is the final goal*?).

After applying the proposed framework to the 47 definitions, we find that 79% contain five or more elements of the definitions (Fig 7), which implies significant robustness of the proposed framework.

Table 2 shows the three primary outcomes for each element and the percentage of definitions containing the term. Not surprisingly, the level of consensus calculated through the HHI varies significantly across concepts. Additionally, for each element, the table contains the percentage of definitions that contain the element.

Starting with the conceptual elements, there are two different perspectives: the *regulatory approach*, where open banking is understood as a legal construct, and the *framework approach*, which focuses on the interactions between players, regardless of the regulation. This duality is

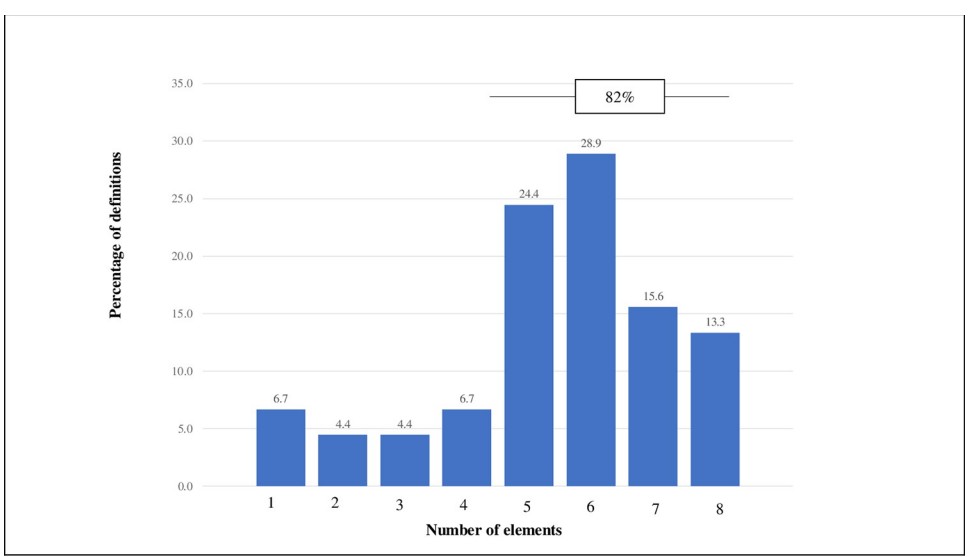

**Fig 7. Completeness of the definitions.**

compatible with the fact that there are specific open banking regulations in some geographical areas (UK, Europe, and Australia). In contrast, in other regions (US and Canada), open banking exists as a phenomenon but without a specific regulation in place yet. We find a tight relationship between *Nature* and *Consent*, considering that regulation implies requirement, obligation, or empowerment, while framework implies enablement.

Regarding core attributes, the main keywords are "sharing" for *Action* and "APIs" for *Process*. Nevertheless, the interpretation of both should be significantly different. Regarding *Action*, there is a high consensus among all definitions around "sharing," which is consubstantial with the very notion of open banking as currently understood by practitioners [47]. However, talking about *Process*, although currently, APIs are the most common system interface

**Table 2. Summary of definitions' descriptive statistics.**

|  | HHI | 1 | 2 | 3 | % Def |
|---|---|---|---|---|---|
| **Nature** | 822.1 | regulation | framework | model | |
| (%) | | 16.1 | 12.9 | 9.7 | 64.6 |
| **Consent** | 1,035.2 | enables | requires | allows | |
| (%) | | 21.9 | 12.5 | 12.5 | 66.7 |
| **Subject** | 2,052.5 | customers | banks | third-parties | |
| (%) | | 30.6 | 25.0 | 19.4 | 75.0 |
| **Action** | 2,281.4 | share | build | release | |
| (%) | | 46.2 | 5.1 | 2.6 | 83.3 |
| **Object** | 348.5 | customer data | data | apps and services | |
| (%) | | 7.7 | 5.1 | 5.1 | 83.3 |
| **Recipient** | 593.1 | 3rd parties | Auth. 3rd partird | fintechs | |
| (%) | | 16.1 | 6.5 | 6.5 | 66.7 |
| **Process** | 3,395.1 | APIs | open APIs | secure APIs | |
| (%) | | 50.0 | 27.8 | 5.6 | 50.0 |
| **Purpose** | 451.4 | n.a. | n.a. | n.a. | |
| (%) | | n.a. | n.a. | n.a. | 37.5 |

technology, the open banking phenomenon could be perfectly conceived by leveraging different interface technologies such as screen scraping [48]. That is why API should be deemed a relevant yet not essential element in the definition of open banking.

As for *Subject*, there is a low degree of consensus: 30.6% definitions are built around "*customer*," 25.0% around "*banks*" (including synonyms such as "*financial institutions*"), and 19.4% around "*third parties*." This lack of convergence emerges from the fact that open banking can be formulated under three perspectives: the client perspective: "*customers–share*," institutional perspective: "*banks–make available*," and ecosystem perspective: "*third parties–access*." However, it is still unclear which approach is better. Nevertheless, the fact is that comparing roles of the three main actors in the open banking process, banks are passive agents, and their only function is to facilitate access to data. Similarly, third parties such as fintechs, for that matter any third party, cannot force a customer to enter into an open banking relationship with a banking client. That is why the client perspective seems crucial to understanding the essence of open banking as a "right to share" rather than a "right to access."

The *Object* of open banking is also unclear, ranging from "*data*" to "*applications and services.*" Lastly, concerning the *Recipient*, there are different levels of concretion, from a general conception ("*third parties*") to specific type players ("*fintechs*"). There is, however, one open matter, "payments initiation." Apart from data sharing, some regulations also include payment initiation as an object of open banking (e.g., UK, EU, India, and Brazil). However, there are minimal academic literature references to this matter. Thus, we will attach to the mainstream definitions of open banking as data sharing.

Finally, the *Purpose* element is highly undefined. Although *"transparency"* and "*competition"* appear in several cases, there is no convergence in the final goal of open banking in any of the analyzed definitions.

In sum, although consensus around different elements of open banking is limited, it could be defined as "a generally regulated framework that enables banking customers to share their data with third parties, commonly through standardized interfaces such as APIs, to increase competition in the financial sector." The proposed definition covers the eight elements identified in the proposed open banking integrated framework and could be understood as a generalization of all the analyzed partial definitions.

**Observation 3.1.**   There is neither a single definition of open banking in the academic literature nor a specific definition by knowledge domain. Instead, there is a collection of idiosyncratic and paper-specific approaches toward its definition.

**Observation 3.2.**   Among existing definitions, there are strong commonalities in some elements, while others show a high degree of dispersion. These differences arise mainly from different knowledge domains through which open banking is analyzed and various jurisdictions where it occurs.

**Observation 3.3.**   Despite underlying divergences, a standard definition of open banking can be formulated and leveraged in all conceptual domains based on the proposed approach.

**Observation 3.4.**   Despite customers playing a central role in different definitions of open banking as the owner of data, decision-maker of data sharing, and target of the framework's purpose, one key element where prior research lacks consensus and focus is the role of a banking customer within open banking. Only 30.6% definitions are built around the word "client" (compared with 25.0% definitions that are built around "banks" and 19.4% around "third parties").

## 5. Discussion and conclusions

Our bibliometric analysis confirms the academic community's limited but growing interest in open banking and the challenges of a multidisciplinary approach to the phenomenon.

Together with the intrinsic fragmentation in the analysis of the phenomenon due to its regulatory facets, both elements result in a corpus of literature that is still getting consolidated but lacks some foundations for further development.

Based on the clustering analysis' results of the nascent literature, four conceptual clusters have been identified. These are (i) the platformization of the retail banking industry business model; (ii) a manifestation of the overall data sharing trend applied to the banking data; (iii) the interaction between the emergent fintech ecosystem and incumbent financial institutions; and (iv) the regulatory framework that, in some jurisdictions, bolsters the open banking phenomenon. These four clusters can be interpreted as different connotations underpinning the concept of "open banking." Hence, the complex nature of open banking is a considerable challenge for future literature development, as partial analysis of the phenomenon will yield limited conclusions. Thus, only multidisciplinary approaches will offer good insights.

A clustering analysis to identify the conceptual domains around the open banking definition is also a valuable contribution. As an unsupervised learning methodology, clustering analysis returns an objective output, eliminating pre-classification biases. Moreover, the clustering approach unveils all the critical factors behind the open banking concept, supporting our proposal of an integrative definition valid across all disciplines and realizations of open banking. Consequently, although there are strong linkages between Cluster 1 (Business model/Platform), Cluster 4 (Regulation), and the academic literature emanating from Business Management and Social Sciences-Law, respectively, Cluster 2 (Data sharing) and Cluster 3 (Fintech) unveil purely transversal conceptual domains, multidisciplinary in nature that do not match with a single academic field and that could not have been identified without the clustering approach.

The detailed analysis of the 47 identified idiosyncratic and working definitions of the phenomenon confirms the need for a generalized conceptualization that amalgamates all existing perspectives on the topic. The proposed framework arising from the definition analysis is by itself a valuable tool for understanding the depth of open banking and the importance of identifying all relevant components that intervene in its dynamics. It is also important to note that the different formulations for the *Subject* of open banking constitute three perspectives of the phenomenon. These include (i) the "institutional perspective," which analyzes open banking based on the obligations to comply with banking regulation; (ii) the "ecosystem perspective," which focuses on the potential mechanics and benefits for new entrants, especially fintechs, from accessing banking clients' data; and (iii) the "client perspective," which studies the fundamental data-sharing right that constitutes the basis of open banking. Although the literature has not been explicit on this matter, researchers need to understand the implications of each positioning.

This study contributes to filling the literature gap with a potential generalized multidisciplinary open banking definition. Our proposed definition encompasses the four conceptual domains identified through the cluster analysis of the existing literature. Further, our proposed definition contributes to synthesizing different approaches, serving as a catalyzer for further research on the topic and significantly enhancing multidisciplinary approaches to the question.

Our proposed generalized definition should help increase collaboration among researchers from different academic disciplines and cooperation among researchers in different geographies to analyze the open banking phenomenon. Additionally, the proposed definition is especially relevant for policymakers and private economic agents, considering current ongoing discussions around the evolution of open banking regulation. Finally, the generalization of the open banking concept is also relevant for end customers as data owners and primary beneficiaries of open banking regulations.

The main limitation of this analysis is the emergent nature of the existing literature. Although several quality filters have been applied to the inputs to ensure the quality of the outcomes, this approach could be replicated in the future on articles published in peer-reviewed journals once a sufficient corpus of high-quality literature has been developed.

## Supporting information

**S1 Annex. Open banking definitions** [2, 5, 9, 13–17, 19, 24, 26, 31, 32, 48–81].
(DOCX)

**S2 Annex. Analytical approach** [82–84].
(DOCX)

**S1 File. Cluster map.**
(TXT)

**S2 File. Cluster network.**
(TXT)

**S3 File. Terms thesaurus.**
(TXT)

**S4 File.**
(RIS)

## Author Contributions

**Conceptualization:** Gorka Koldobika Briones de Araluze, Natalia Cassinello Plaza.

**Data curation:** Gorka Koldobika Briones de Araluze.

**Formal analysis:** Gorka Koldobika Briones de Araluze.

**Funding acquisition:** Gorka Koldobika Briones de Araluze, Natalia Cassinello Plaza.

**Investigation:** Gorka Koldobika Briones de Araluze, Natalia Cassinello Plaza.

**Methodology:** Gorka Koldobika Briones de Araluze, Natalia Cassinello Plaza.

**Project administration:** Gorka Koldobika Briones de Araluze.

**Resources:** Gorka Koldobika Briones de Araluze.

**Software:** Gorka Koldobika Briones de Araluze.

**Supervision:** Gorka Koldobika Briones de Araluze, Natalia Cassinello Plaza.

**Validation:** Gorka Koldobika Briones de Araluze, Natalia Cassinello Plaza.

**Visualization:** Gorka Koldobika Briones de Araluze.

**Writing – original draft:** Gorka Koldobika Briones de Araluze, Natalia Cassinello Plaza.

**Writing – review & editing:** Gorka Koldobika Briones de Araluze, Natalia Cassinello Plaza.

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
