## [Decision Letter · Decision Letter 0]

21 Mar 2022

PONE-D-22-03690Open banking: A bibliometric analysis-driven definitionPLOS ONE

Dear Dr. Briones de Araluze,

Thank you for submitting your manuscript to PLOS ONE. After careful consideration, we feel that it has merit but does not fully meet PLOS ONE’s publication criteria as it currently stands. Therefore, we invite you to submit a revised version of the manuscript that addresses the points raised during the review process.

We look forward to receiving your revised manuscript.

Kind regards,

Tawei (David) Wang

Academic Editor

PLOS ONE

Journal Requirements:

2. We note you have included a table to which you do not refer in the text of your manuscript. Please ensure that you refer to Table 2, 3, 4, 5, and 6 in your text; if accepted, production will need this reference to link the reader to the Table.

Reviewers' comments:

Reviewer's Responses to Questions

**Comments to the Author**

1. Is the manuscript technically sound, and do the data support the conclusions?

Reviewer #1: Yes

Reviewer #2: Yes

2. Has the statistical analysis been performed appropriately and rigorously? 

Reviewer #1: Yes

Reviewer #2: Yes

3. Have the authors made all data underlying the findings in their manuscript fully available?

Reviewer #1: Yes

Reviewer #2: Yes

4. Is the manuscript presented in an intelligible fashion and written in standard English?

Reviewer #1: Yes

Reviewer #2: Yes

5. Review Comments to the Author

Reviewer #1: As a relatively new concept, open banking has generally been considered as the transformational lever of the retail banking industry. However, its theoretical conceptualization was underdeveloped in the literature. This study presents a potential generalized definition of open banking based on bibliometric and discourse analysis. This paper is interesting and provides timely insights into the literature. Please refer to my review report for details.

Reviewer #2: Review report titled: Open Banking: A bibliometric analysis-driven definition

Summary of the paper:

By using bibliometric and discourse analysis, this paper proposes a standard and multidisciplinary definition integrating business model change, client data sharing, incorporation of technological companies (fintechs and others), and regulation. There are two major contributions. First, these four elements foresee open banking as a critical enabler of a new strategic dynamic in banking, which should also be considered in any new regulation in a globalized banking sector. Secondly, authors find a substantial literature gap regarding the role of the banking client in the open banking dynamics.

Major concerns:

1. Motivation: this paper needs to find additional motivations and implications. Consider, for example, why integrating the definition of open banking is essential for practitioners and scholars.

2. Literature review: the literature review section is too long. It may be helpful for the authors to reorganize this part, especially to articulate the literature gap and what this paper adds to it. In other words, author may combine section 2.1 Literature review and section 2.2 research question.

3. The methodology section: in this paper does not use self-developed methodology, so section 3, especially section 3.1, is way too long and gives too much background information. Maybe the author could put some background information in the appendix instead. Additionally, I am not sure what is the advantage of the two methodologies authors applied and why they fit into the research question. This part of the paper should be emphasized more than introducing the background knowledge.

Minor concerns:

1. Page 7, line 10 “Existing literature can be classified under four different perspectives: regulatory, technological, economic, and managerial.” doesn’t need to a separate paragraph.

2. Notions should keep consistent. Page 27, line 16 “Third, Open Banking enhances the entrance to the banking market of non-banking players…” Please make sure the notions are consistent, i.e., whether use capital letters.

3. The section 5. Discussion and conclusions could be shortened. The current version of section 5 is too long, and repeated information is mentioned.

6. PLOS authors have the option to publish the peer review history of their article (what does this mean?). If published, this will include your full peer review and any attached files.

Reviewer #1: No

Reviewer #2: **Yes: **Ji Sui

---

## [Author Response · Author response to Decision Letter 0]

23 May 2022

All relevant responses to editor and reviewers have been included in the "Response to Reviewers" attached word document

---

## [Decision Letter · Decision Letter 1]

1 Jul 2022

PONE-D-22-03690R1Open banking: A bibliometric analysis-driven definitionPLOS ONE

Dear Dr. Briones de Araluze,

Thank you for submitting your manuscript to PLOS ONE. One of the reviewers still have some minor comments which I believe will make the paper better.

We look forward to receiving your revised manuscript.

Kind regards,

Tawei (David) Wang

Academic Editor

PLOS ONE

Journal Requirements:

Reviewers' comments:

Reviewer's Responses to Questions

**Comments to the Author**

1. If the authors have adequately addressed your comments raised in a previous round of review and you feel that this manuscript is now acceptable for publication, you may indicate that here to bypass the “Comments to the Author” section, enter your conflict of interest statement in the “Confidential to Editor” section, and submit your "Accept" recommendation.

Reviewer #1: (No Response)

Reviewer #2: All comments have been addressed

2. Is the manuscript technically sound, and do the data support the conclusions?

Reviewer #1: Yes

Reviewer #2: Yes

3. Has the statistical analysis been performed appropriately and rigorously? 

Reviewer #1: N/A

Reviewer #2: Yes

4. Have the authors made all data underlying the findings in their manuscript fully available?

Reviewer #1: Yes

Reviewer #2: Yes

5. Is the manuscript presented in an intelligible fashion and written in standard English?

Reviewer #1: No

Reviewer #2: Yes

6. Review Comments to the Author

Reviewer #1: I appreciate the authors’ responses to my previous review comments. Overall, the revisions have improved the paper, and I continue to find the paper interesting. Although I am pleased that these efforts have addressed some of my comments, a few concerns remain unaddressed. Please see attached review report for detail.

Reviewer #2: I have raised three major concerns in the first round of the review process.

The first relates to motivation and implications. The authors did a good job of emphasizing the motivation and articulating implications from multiple perspectives, e.g., institutional, ecosystem, etc. Secondly, in this version, the authors offer detailed citations and quotations to highlight the literature gap, thereby addressing my second concern. Finally, the authors add an appendix as I recommended, which clarifies the main body of the paper.

Overall, this version is well written and has a decent structure. In my opinion, this paper should be accepted.

Authors may accidentally submit their draft with the changes track at the back of the document. I am not sure if this is a required feature or a mistake.

7. PLOS authors have the option to publish the peer review history of their article (what does this mean?). If published, this will include your full peer review and any attached files.

Reviewer #1: No

Reviewer #2: **Yes: **Ji Sui

---

## [Author Response · Author response to Decision Letter 1]

3 Aug 2022

A rebuttal letter that responds to each point raised by the academic editor and reviewer has been included with the submission.

---

## [Editor Report · Decision Letter 2]

24 Aug 2022

PONE-D-22-03690R2Open banking: A bibliometric analysis-driven definitionPLOS ONE

Dear Dr. Briones de Araluze,

Thank you for submitting your manuscript to PLOS ONE. After reading through the submission, I believe the authors have carefully and effectively, to some extent, addressed the reviewers' comments and suggestions so I  decided not to send it out for review again. However, before I can formally accept the paper, can the authors send the manuscript to a professional copy editor for another round of proofreading? The authors can provide the journal's author guide so the copy editor can help the authors fix most of the writing issues and formatting issues. 

We look forward to receiving your revised manuscript.

Kind regards,

Tawei (David) Wang

Academic Editor

PLOS ONE
---

## [Author Response · Author response to Decision Letter 2]

17 Sep 2022

Based on your suggestion, we have got our manuscript proof-read from a professional copy-editor. The manuscript has been rechecked and the necessary changes have been made in accordance with the reviewers’ suggestions. 

As per editor’s comments, all the references have been doubled checked. During the process, we realized that one article was retracted during the submission process: 

Al-Rfouh, A. (2019). Assessing Banks to Develop an Open Digital Marketing Approach for the Future of the Bank. Electronic Business Journal, 18(3), 20-31.

As we have not found the retraction notice, the article has been removed from the references, and the article has been updated accordingly, implying non-significant changes in the calculations of the percentages contained in Table 2 (Summary of definitions' descriptive statistics).

---

## [Editor Report · Decision Letter 3]

19 Sep 2022

Open banking: A bibliometric analysis-driven definition

PONE-D-22-03690R3

Dear Dr. Briones de Araluze,

We’re pleased to inform you that your manuscript has been judged scientifically suitable for publication and will be formally accepted for publication once it meets all outstanding technical requirements.

Kind regards,

Tawei (David) Wang

Academic Editor

PLOS ONE
---

## [Editor Report · Acceptance letter]

21 Sep 2022

PONE-D-22-03690R3 

Open banking: A bibliometric analysis-driven definition 

Dear Dr. Briones de Araluze:

I'm pleased to inform you that your manuscript has been deemed suitable for publication in PLOS ONE. Congratulations! Your manuscript is now with our production department. 

Kind regards, 

on behalf of

Dr. Tawei (David) Wang 

Academic Editor

PLOS ONE